# Exosomes combined with extra virgin olive oil reduces lipogenesis, oxidative stress, and inflammation in non-alcoholic fatty liver disease model

Bassam F. Alowaiesh[1,2]*, Doaa Ibrahim[3], Ayman A. Saleh[4], Haifa A. S. Alhaithloul[1,2], Abdulsalam A. M. Alkhaldi[1,2], Ahmed Abdelfattah-Hassan[5,6]*

1 Olive Research Center, Jouf University, Sakaka, Saudi Arabia, 2 Biology Department, College of Science, Jouf University, Sakaka, Saudi Arabia, 3 Department of Nutrition and Clinical Nutrition, Faculty of Veterinary Medicine, Zagazig University, Zagazig, Egypt, 4 Department of Animal Wealth Development, Genetics & Genetic Engineering, Faculty of Veterinary Medicine, Zagazig University, Zagazig, Egypt, 5 Department of Veterinary Medicine, College of Applied and Health Sciences, A'Sharqiyah University, Ibra, Sultanate of Oman, 6 Department of Anatomy and Embryology, Faculty of Veterinary Medicine, Zagazig University, Zagazig, Egypt

* bfalawish@ju.edu.sa (BFA); aabdelfattah@vet.zu.edu.eg, abdelfattah@zewailcity.edu.eg (AA-H)

## Abstract

In response to high-fat-diet, excessive lipid accumulation in the liver results in chronic damage and inflammation. Olive oil has been studied for its health beneficial effects in hyperlipidemia (mainly has lipids lowering and antioxidative potential) while mesenchymal stem cells derived exosomes (MSCs-Exo) are investigated mainly for their tissue regenerative and anti-inflammatory potential. In the present study we aimed to combine the beneficial effects of Extra Virgin Olive Oil (EVOO) and MSCs-Exo on a model of high-fat-diet induced Non-Alcoholic Fatty Liver Disease (NAFLD, which still lacks effective treatment protocols) and detect whether an improved response could be achieved from this combination. Sprague Dawley rats (n = 40) were randomly assigned to five groups (n = 8/group), control, hyperlipidemia (HL), HL+EVOO, HL + Exo and HL + Exo+EVOO. Our results show that better antihyperlipidemic effects were obtained in the combined group receiving Exo+EVOO treatment more than using EVOO or MSCs-Exo alone. This was achieved by improving plasma lipids profile, improving antioxidants stores and reducing lipid peroxidation, no change in liver function parameters which was confirmed also by the histopathological examination of the liver where a preserved normal liver architecture. To further elucidate the mechanisms involved, the gene expression levels of lipogenesis (SREBP-1c, ACC, FAS, GPAT3, SCD1, and FSP27), inflammation (IL-1β, TNF-α, IL-6, IL-18, CCL20, and NF-κB), lipid peroxidation (CPT1A, ACOX1) and PPAR pathway (PPARα, PPARγ) were all normalized. This indicates that combined Exo+EVOO harnessed the benefits of both, and this was much better in treating hyperlipidemia and NAFLD and warrants prospects for approaches that could be adopted to treat NAFLD.

**Data availability statement:** All data are contained within the manuscript.

**Funding:** This work was funded by the Deanship of Scientific Research in cooperation with Olive Research Center at Jouf University under Grant Number (DSR2022-RG-0164). The funder had no role in study design, data collection and analysis, decision to publish, or preparation of the manuscript.

**Competing interests:** The authors have declared that no competing interests exist.

## Introduction

Non-alcoholic fatty liver disease (NAFLD) is defined as steatosis or accumulation of fat exceeding 5% of the hepatocytes in the absence of significant alcohol intake [1]. It is estimated that up to 30% of the population in western countries and almost 25% worldwide [2]. Having excess fat accumulation in the liver and a degree of NALFD. In general, impaired or dysfunctional lipid metabolism is associated with a pro-inflammatory state and increased risk of atherosclerosis and cardiovascular diseases [3,4] and are one of the major risk factors for early death in humans. Also, the occurrence of oxidative stress exacerbates the disease condition in patients with hyperlipidemia [5]. Histologically (using liver biopsy), NAFLD can range from simple steatosis to severe non-alcoholic steatohepatitis (NASH); the latter is a combination of steatosis and various degrees of inflammation and fibrosis. Mechanistically, hyperlipidemia is closely associated with oxidative stress due to increased production of oxygen free radicles, defective antioxidant system and increased inflammatory state [6–8]. Consequently, there is a significant increase in lipid peroxidation products and a decrease in antioxidant levels in the plasma of hypercholesterolemic patients [8].

Olive oil, which is a major constituent of the Mediterranean diet, is widely accepted as a beneficial functional food because it has a high content of mono-unsaturated fatty acids (MUFAs) in addition to other beneficial components such as polyphenols and tocopherols, and, to a lesser extent, some vitamins [9]. Studies have also found that polyphenols are present in higher concentration in extra virgin olive oil (EVOO) than in ordinary refined olive oil, giving EVOO an advantageous quality. In addition to its monounsaturated fatty acid content and antioxidant activity, the unique health benefits of extra virgin olive oil are attributed to a rich composition of minor compounds, including phenolic compounds (e.g., oleuropein, hydroxytyrosol), secoiridoids (e.g., oleocanthal), and phytosterols. These constituents contribute independently and synergistically to EVOO's anti-inflammatory, antioxidant, and metabolic effects, and are key differentiators from refined oils or other dietary fats. Olive oil has been intensively studied for various beneficial effects on human health, such as protective cardiovascular properties, modulator of lipid metabolism, preserving antioxidant stores, has anti-inflammatory properties, has anti-tumor properties and many other beneficial biological functions [10]. While extra virgin olive oil is valued for its rich content of monounsaturated fats and bioactive compounds, including polyphenols, it is important to note that its minimally processed nature retains components sometimes referred to as "impurities." These include various phenolic and volatile compounds. Although these components are generally beneficial and contribute to EVOO's antioxidant and anti-inflammatory properties, in very high concentrations, certain phenolics may exhibit mild anti-nutritional effects by interfering with nutrient absorption. However, such effects are not typically observed at dietary intake levels and do not outweigh the overall health benefits associated with EVOO consumption [10]. Previous studies in animal models and in humans have shown that olive oil administration has various health-beneficial effects preventing the development and/or progression of NAFLD [11–16].

Mesenchymal stem cells (MSCs) represent a heterogeneous cell population residing in various adult tissues, such as bone marrow, peripheral blood, umbilical cord, adipose tissue and other tissues. These cells are in the spotlight of regenerative therapy-research due to their various beneficial effects on many diseases and in regenerative medicine. One of the characteristics of stem cells in general, and bone marrow-derived MSCs (BM-MSCs) as well, is that it can be converted to many other cell types including osteoblasts or adipocytes, which is one of the features used to confirm their stemness [17–20]. However, such adaptogenic power of stem cells could contribute more to advancing NAFLD especially that liver stem cells were shown to be involved in the progression of NAFLD [21,22], because of NAFLD on liver microenvironment. Also, the deregulation of liver stem cells can lead to increased fibrosis and stage progression of liver diseases [23]. In this regard, few studies in the literature were found and these provided scarce information on the use of stem cells for the treatment of hyperlipidemia/NAFLD.

Exosomes are a type of nanovesicles secreted by body cells, and they are key components for the intercellular communication process. Exosomes derived from stem cells are similar if not superior to their cells of origin in their therapeutic promise, due to their cargo of various important mRNA, micro-RNA, and other factors [24,25]. More studies were found in the literature on the use of exosomes derived from stem cells than there was on the use of stem cells alone or combined with other therapeutics such as Liraglutide on the treatment of liver lesions in NAFLD [26,27]. The exact reason for this is not clear, but it could be explained as liver NAFLD microenvironment's effect, which was shown to be involved in the deregulation of endogenous liver stem cells, could also deregulate exogenously administered stem cells [23]. In direct relation to this, it was shown that the first two years after allogenic hematopoietic stem cells transplantation were associated with the development of hypercholesterolemia and hypertriglyceridemia in approximately 73% of treated cases and 29% of these patients received statins treatment as lipid lowering therapy [28]. Although stem cells represent a promising cell-based therapy for various disease conditions, their use in treating NAFLD seems controversial. However, stem cells-derived exotics should be a better alternative, since these exosomes cannot convert into other cell types and their cargo is dependent on the condition of the original cells rather than the microenvironment which exogenous stem cells might encounter during NAFLD.

Recently, a study highlighted several differentially expressed genes that are key for the development of NAFLD and are related to inflammatory reaction and lipid synthesis within the liver and can lead to the progression towards severe NASH and eventually the development of hepatocellular carcinoma [29]. These key genes include genes related to lipid accumulation, lipid peroxidation and inflammation, such as Fatty Acid Binding Protein 5 (FABP5), Stearoyl-CoA Desaturase (SCD), C-C Motif Chemokine Ligand 20 (CCL20), Glycerol-3-Phosphate Acyltransferase 3 (AGPAT9 or GPAT3), Perilipin 1 (PLIN1), and Interleukin 1 Receptor Antagonist (IL1RN), highlighting the complex etiology of this disease. There is no data in the literature on the effect of the combination of extra virgin olive oil with stem cells-derived factors on treating hyperlipidemia induced NAFLD nor on these key NAFLD genes. Therefore, we aimed to evaluate whether the combined use of extra virgin olive oil and MSCs-exosomes will be more beneficial than using either alone, for treating the inflammation and dyslipidemia caused by NAFLD with special focus on the key genes implicated in the lipogenesis, lipid oxidation and inflammation pathways.

## Materials and methods

### Animals

Forty apparently healthy male Sprague Dawley rats were purchased from the animal house at the Faculty of Veterinary medicine, Zagazig University. The rats weighed about 180 ± 22g at purchase and were accommodated for 2 weeks before any experimental approach was performed. The rats were housed (4 rats/cage) in standard lab conditions at a temperature of 24 ± 2°C, relative humidity 50:55% and were kept at 12 hours light/dark cycle. The rats were freely offered a standard rat chow diet with *ad libitum* access to drinking water. The study was conducted according to relevant regulations,

including ARRIVE Guidelines (2.0), and the study was conducted in strict accordance with the recommendations in the Guide for the Care and Use of Laboratory Animals of the National Institutes of Health (NRC, Washington D.C., USA). The experimental protocol was approved by the Institutional Animal Care and Use Committee of Zagazig University (protocol number: ZU-IACUC/2/F/373/2023).

## Experimental design

After accommodation, the rats were weighted and were randomly assigned to one of the following study groups (n = 8/group): healthy control group (Control) which was fed only standard rat chow diet, hyperlipidemia group (HL) fed high fat diet only, hyperlipidemia group fed high fat diet and treated with oral gavage of 0.5 ml extra virgin olive oil daily (HL+EVOO), hyperlipidemia group fed high fat diet and treated with 50 μg exosomes derived from mesenchymal stem cells once/week (HL + Exo), hyperlipidemia group received high fat diet and treated with both exosomes and extra virgin olive oil (HL + Exo+EVOO). The study continued for 8 weeks. The high fat diet consisted of 80% standard rat chow diet plus 18% plant-oil based margarine (a commercial margarine available in the market) and 2% cholesterol (Sigma Aldrich) [30]. Extra virgin olive oil was provided from Olive Oil Research Unit, Al-Jouf University, Al-Jouf, Saudi Arabia. Bone marrow mesenchymal stem cells (BM-MSCs) isolation and characterization was as we previously reported [31,32], and exosomes obtained from BM-MSCs were isolated and characterized as we previously performed and were administered once weekly by intraperitoneal injection of 50 μg of exosomes in 200 μL of PBS. Exosomes derived from BM-MSCs were administered intraperitoneally at a dose of 50 μg per week, based on total protein content, as commonly reported in preclinical studies. The i.p. route was chosen for its practicality, safety, and demonstrated efficacy in delivering exosomes systemically in small animal models. Exosomes derived from BM-MSCs were administered intraperitoneally (i.p.) at a dose of 50 μg total exosomal protein per rat in 200 μL PBS, given once weekly for the duration of the study. The dose was selected based on previously reported preclinical dose–response and efficacy studies in rats that have used protein-based dosing (many groups quantify EV dose by total protein and report therapeutic effects in the 20–400 μg/animal range depending on route and model). The i.p. route was chosen for practicality and reproducibility in small animals and because i.p. administration results in systemic exposure via lymphatic absorption and subsequent systemic distribution. [33]. At the end of the study, and before samples' collection, the rats were weighed and then euthanasia was performed using isoflurane in a closed chamber according to American Veterinary Medical Association (AVMA) Guidelines for the Euthanasia of Animals (2020). Blood was immediately collected by cardiac puncture for lipid profile and liver function parameters and rats' abdomen were opened and the intact liver was exercised and weighed. The rats' blood was collected by cardiac puncture and the obtained blood samples were kept in anticoagulant-free tubes and allowed to clot for 30 min. After that, the tubes were centrifuged at 350$g$ for 15 min. The serum was collected into 1.5 ml Eppendorf tubes and stored at −20°C for subsequent biochemical analyses.

## Lipid profile assessment

The assessment of lipid profile: including triglycerides (TG), total cholesterol (TC), high-density lipoprotein (HDL-C), and low-density lipoprotein (LDL-C), all were performed using standard biochemical procedures and using commercially available kits (from Bio-diagnostics Co., Cairo, Egypt), following the manufacturer's instructions and in triplicates.

## Measurement of antioxidant parameters

The serum level of malondialdehyde (MDA) was assessed as an indicator for the state of lipid peroxidation. While the total antioxidant capacity (TAC) and levels of glutathione (GSH) were evaluated as indicators for the antioxidant stores in the study rats. The levels of TAC, GSH and MDA were measured using available kits (Bio-diagnostics Co., Cairo, Egypt and BioVision, Inc., Milpitas, CA, USA), following manufacturer's instructions and in triplicates.

### Measurement of liver function parameters

The liver function parameters, such as aspartate transaminase (AST) and alanine aminotransferase (ALT), total proteins and albumin in serum samples were evaluated. These were tested in triplicate using commercially available kits and following manufacturer's protocol.

### Histopathological evaluations

After weighing the intact livers, a piece of the liver tissue was obtained and immediately fixed in freshly prepared 10% neutral buffered formalin solution for 24 hours. After that the liver specimen was histologically processed using standard histopathological techniques (dehydration, embedding in paraffin) as we previously reported [34]. The liver paraffin blocks were cut using microtome into 5µm sections which were stained with standard H&E staining procedure. The histopathological examination of the liver sections from different groups was blindly assessed by an experienced pathologist.

### Gene expression analyses

Liver specimens were immediately collected post sacrifice, and the samples were immediately immersed in Qiazol Lysis Reagent (Cat. No.: 79306, Qiagen, USA) and then quickly stored at −20°C. After that, total RNA extraction was performed using RNA extraction kit (RNeasy Mini Kit, Cat. No.: 74106, Qiagen, USA) following manufacturer instructions. The concentration and purity of obtained total RNA was analyzed using nanodrop at 260 and 280 nm wavelengths (Quawell Q5000, Quawell Technology, Inc., San Jose, CA, USA). Reverse transcription of the obtained mRNA into cDNA was done using the RevertAid First Strand cDNA Synthesis Kit (Thermo Scientific, Cat. No. 1622), following the kit's instructions. The primer sequences for the studied genes are shown in Table 1, the expression levels were analyzed using StepOnePlus™ Real-Time PCR system (Applied Biosystems, Waltham, MA, USA) by using QuantiTect SYBR® Green PCR Kit (Qiagen, Cat. No. 204141), following manufacturer's instructions. The gene expression levels were normalized against a housekeeping gene (GABDH), and the amplified products were relatively quantified using the $^{-2\Delta\Delta}$Ct method.

### Statistical analysis

The obtained data from this study was statistically analyzed using one-way ANOVA through PASW statistical package (SPSS v18, SPSS Inc., Chicago, IL, USA), followed by Tukey's HSD to show between group differences. The gene expression data was analyzed using one-way ANOVA through GraphPad Prism 5 (GraphPad Software Inc., La Jolla, CA, USA). Statistical significance was considered when P value ≤ 0.05, and the data presented are means ± SD, unless otherwise stated.

## Results

### Rat's body weight and liver/rat weight ratio

Compared to the control group the rats fed high fat diet in the hyperlipidemia group showed significant increase in their body weight, liver weight and in liver to rat weight ratio (Table 2). The administration of olive oil, exosomes or a combination of both to rats fed high fat diet resulted in a significantly lower rats' weight at the end of the study and decreased liver weight and liver/rat weight percent as well. However, there was no significant difference between olive oil, exosomes or olive oil + exosomes groups.

### Plasma lipid parameters

In the hyperlipidemic rats fed on a high fat diet, the plasma levels of triglycerides, total cholesterol and low-density lipoproteins (LDL) were all significantly increased compared to the normal control rats (Fig. 1A, 1B and 1C, respectively), while the levels of high-density lipoproteins (HDL) were significantly lower than the normal control group fed normal diet

**Table 1. Showing the primer sequence for the studied genes.**

| Target gene | Primer sequence (5′-3′) | Accession No. |
|---|---|---|
| SREBP-1c | F- GAGTGCGCAGGAGATGCTAT<br>R- GACTGAAGCTGGTGACTGCT | NM_011480 |
| SCD-1 | F- CACCTGCCTCTTCGGGATTT<br>R- CTTTGACAGCCGGGTGTTTG | NM_009127.4 |
| FAS | F- CAAGTGTCCACCAACAAGCG<br>R- GGAGCGCAGGATAGACTCAC | NM_007988.3 |
| ACC | F- CCACATGACCCAGCACATCT<br>R- ATCGATGGACTTGCGTCTCC | NM_133360 |
| GPAT3 | F- TCCTTTTACCCTCGGCCTTC<br>R- AGAGCTCGAAGTCCCTTCCT | XM_031336995.1 |
| FSP27 | F- GTGTTA GCA CCG CAG ATC G<br>R- CAC GAT TGT GCC ATC TTC C | XM_032905931.1 |
| PPARα | F- ACGATGCTGTCCTCCTTGATG<br>R- GCGTCTGACTCGGTCTTCTTG | NM_001354666.3 |
| PPARγ | F- TGAAGGCTCATATCTGTCTCCG<br>R- CATCGAGGACATCCAAGACAAC | NM_013124.3 |
| CPT1A | F- CTCCGCCTGAGCCATGAAG<br>R- CACCAGTGATGATGCCATTCT | XM_057779279.1 |
| ACOX1 | F- TTATGCGCAGACAGAGATGG<br>R- AGGCATGTAACCCGTAGCAC | NM_001414015.1 |
| NF-κB | F- GAGCTGGTGGAGGCCCTG<br>R- GACAGCGGCGTGGAGAC | NM_001276711.1 |
| IL-1β | F- TGACAGACCCCAAAAGATTAAGG<br>R- CTCATCTGGACAGCCCAAGTC | NM_031512.2 |
| IL-6 | F- CCACCAGGAACGAAAGTCAAC<br>R- TTGCGGAGAGAAACTTCATAGCT | NM_012589.2 |
| IL-18 | F- ATGGCTGCCATGTCAGAAGA<br>R-TTGTTAAGCTTATAAATCATGCGGCCTCAGG | XM_039080945.1 |
| IL1RN | F- AAATCTGCTGGGGACCCTAC<br>R- TCTTCTAGTTTGATATTTGGTCCTTG | XM_021155599.2 |
| TNFα | F- CAGCCGATTTGCCATTTCA<br>R- AGGGCTCTTGATGGCAGAGA | L19123.1 |
| CCL20 | F- GTGGGTTTCACAAGACAGATG<br>R- TTTTCACCCAGTTCTGCTTTG | XM_021174464.1 |
| β-actin | F- CGCAGTTGGTTGGAGCAAA<br>R- ACAATCAAAGTCCTCAGCCACAT | V01217.1 |
| GAPDH | F- TGCTGGTGCTGAGTATGTCG<br>R- TTGAGAGCAATGCCAGCC | NM_017008 |

*SREBP-1c*: sterol regulatory element binding transcription factor 1, *SCD-1*: stearoyl-Coenzyme A desaturase 1, *FAS*: Fatty acid synthase, *ACC*: Acetyl coenzyme A Carboxylase, *GPAT3*: glycerol-3-phosphate acyltransferase 3, *FSP27*: fat-specific protein 27, *PPARα*: Peroxisome proliferator activated receptor alpha. *PPARγ*: Peroxisome proliferator activated receptor gamma, *CPT1A*: carnitine palmitoyl transferase IA, *ACOX1*: Peroxisomal acyl-coenzyme A oxidase 1, *NF-κB*: Nuclear factor kappa B, Interleukin (IL)- 1β, IL-6, IL-18, *IL1RN*: Interleukin-1 receptor antagonist, *TNFα*: tumor necrosis factor α, *CCL20*: C-C motif chemokine ligand 20, *β-actin*: Actin beta, *GAPDH*: glyceraldehyde-3-phosphate dehydrogenase.

(Fig. 1D). Following the administration of olive oil, exosomes or olive oil+exosomes, the treated rats showed significantly lower levels of triglycerides approaching levels similar to the normal control group, with no difference between the three treatments. However, total cholesterol and LDL were reduced to near normal levels in the hyperlipidemia groups treated

**Table 2. Showing the average start weight, average end weight, average end liver weight and liver/rat weight percentage of the study groups.**

| Groups | Initial weight (g) | Final weight (g) | liver weight (g) | Liver/Rat weight (%) |
|---|---|---|---|---|
| Control | 268.38±5.4 | 354.88±12.2[c] | 12.38±1.19[c] | 3.49 |
| HL | 263.13±6.7 | 483.75±17.2[a] | 21.13±1.96[a] | 4.37 |
| HL+EVOO | 267.5±7.4 | 369.625±14.9[b] | 14.37±0.74[b] | 3.89 |
| HL+Exo | 263.5±7.8 | 374.25±15.1[b] | 13.87±1.13[b] | 3.71 |
| HL+Exo+EVOO | 263.5±8.4 | 377.75±9.5[b] | 14.75±1.28[b] | 3.90 |

Bars with different lowercase letters (e.g., a, b, c) indicate statistically significant differences between groups (P<0.05), as determined by Tukey's HSD post hoc test following one-way ANOVA. Groups sharing the same letter are not significantly different.. Hyperlipidemia group (HL), Hyperlipidemia+Olive oil group (HL+EVOO), Hyperlipidemia+Exosomes group (HL+Exo), Hyperlipidemia+Exosomes+Olive oil group (HL+Exo+EVOO).

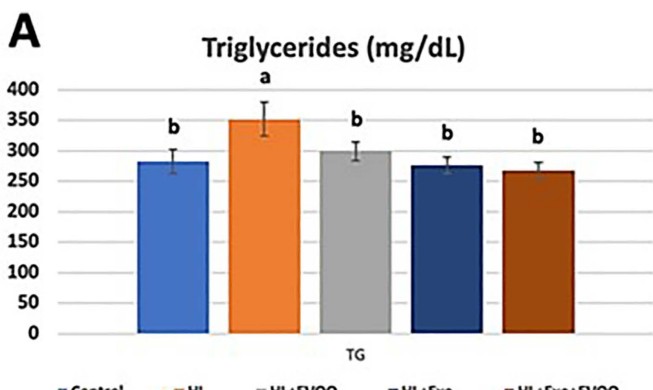
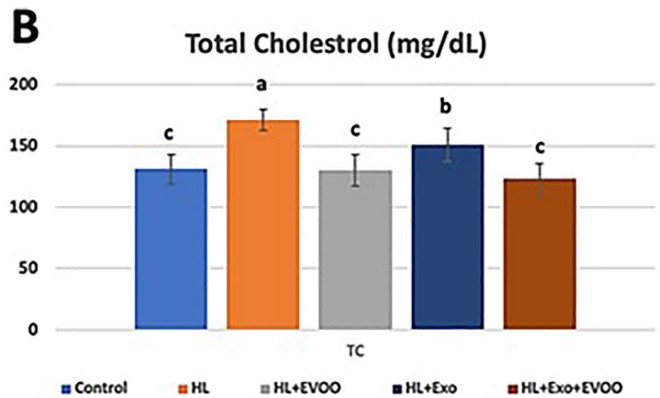
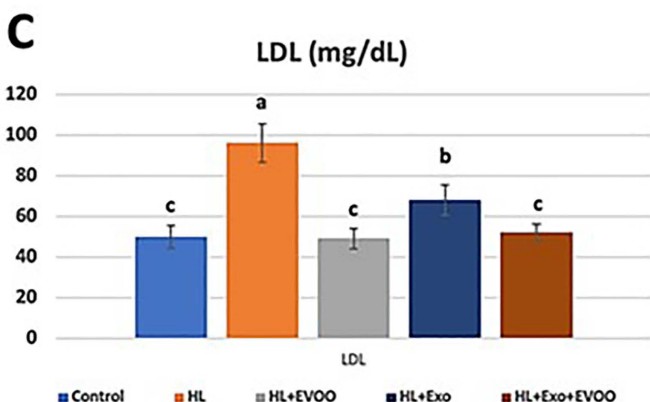
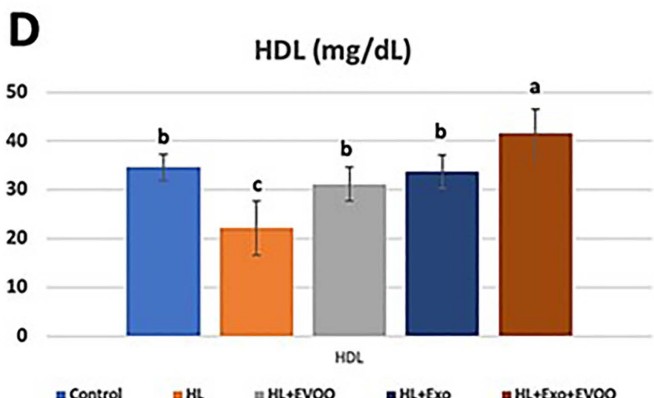

**Fig 1. The plasma lipid parameters of different study groups. (A)** Triglycerides (TGs, mg/dL), **(B)** Total cholesterol (TC, mg/dL), **(C)** Low-density Lipoprotein cholesterol (LDL-C, mg/dL), and **(D)** High-density Lipoprotein cholesterol (HDL-C, mg/dL).

with olive oil or olive oil+exosomes, which was not different from the normal control group. The hyperlipidemic group treated with exosomes only showed moderate reduction in the levels of total cholesterol and LDL which were significantly lower than the hyperlipidemia group but still significantly higher than the normal control group. Also, the HDL levels were restored to near normal levels in the olive oil or exosomes only groups, while its levels were much higher in the group treated with both olive oil+exosomes, compared to other treated or normal control groups (Fig. 1D).

## Antioxidant parameters

High fat diet fed rats showed lower total antioxidant capacity compared to the control group rats and was also significantly lower than the treated groups (Fig. 2A). Also, hyperlipidemia rats showed significant higher malondialdehyde (MDA) levels compared to the other groups and significant lower levels of glutathione (GSH) compared to the remaining groups (Fig. 2B, 2C, respectively). After the administration of olive oil or exosomes to high fat diet fed rats, the levels of MDA and GSH and total antioxidant capacity were restored to near normal levels and were not significantly different from the control group. While the hyperlipidemia group treated with both olive oil and exosomes showed significantly higher levels of glutathione and total antioxidant capacity, even higher than the control non-hyperlipidemic group.

## Liver function parameters

The levels of total protein and albumin in the hyperlipidemia group were slightly higher but insignificantly different from the treated or control groups (Fig. 3A & 3B). However, the levels of ALT and AST were significantly higher in the hyperlipidemia rats fed on a high fat diet compared to the remaining groups (Fig. 3C & 3D). Levels of ALT and AST were reduced in

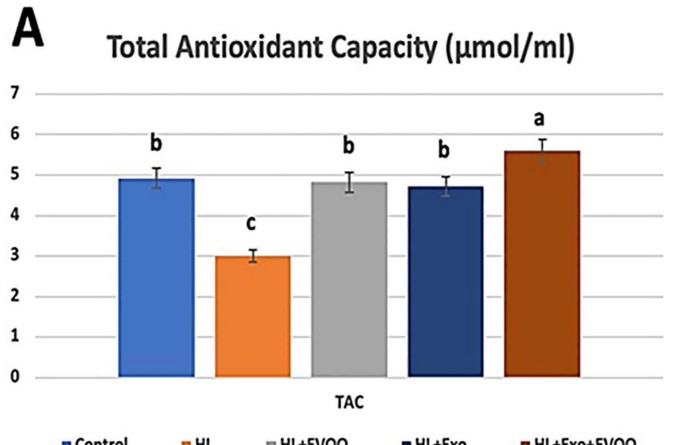

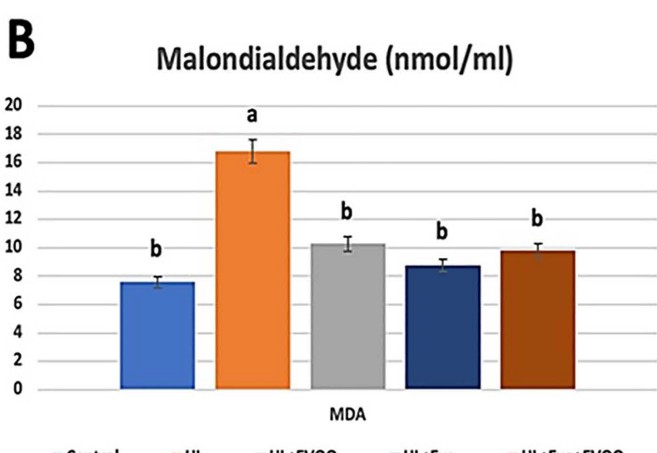

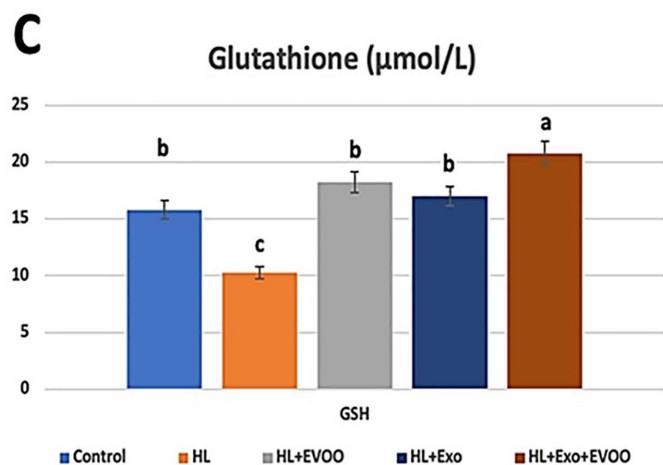

**Fig 2. The plasma antioxidant parameters of different study groups. (A)** Total antioxidant capacity (TAC, μmol/mL), **(B)** Malondialdehyde (MDA, nmol/mL), and **(C)** Glutathione (GSH, μmol/mL).

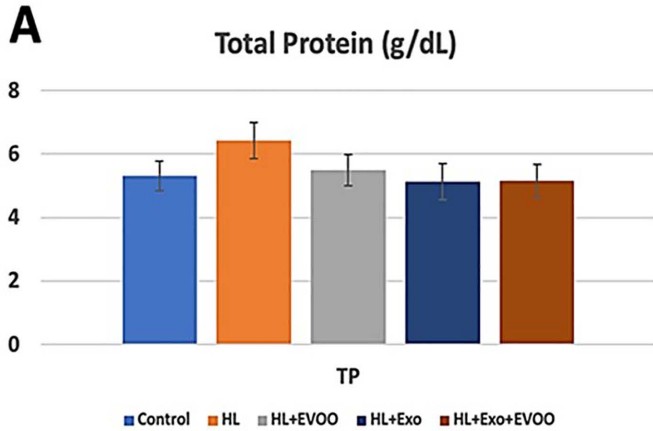

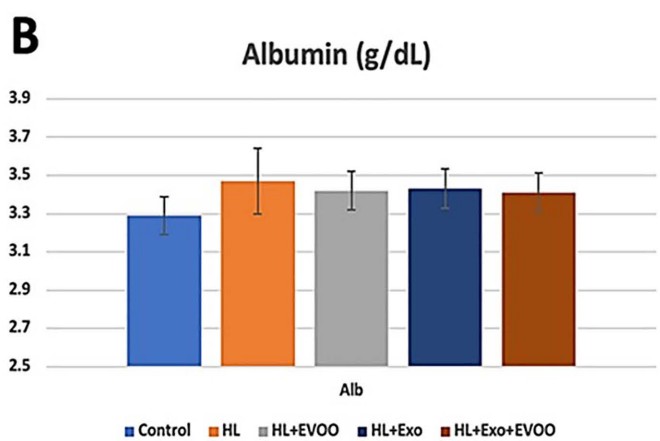

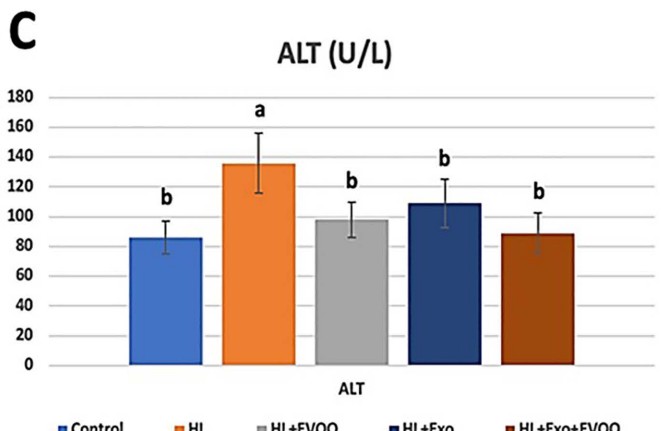

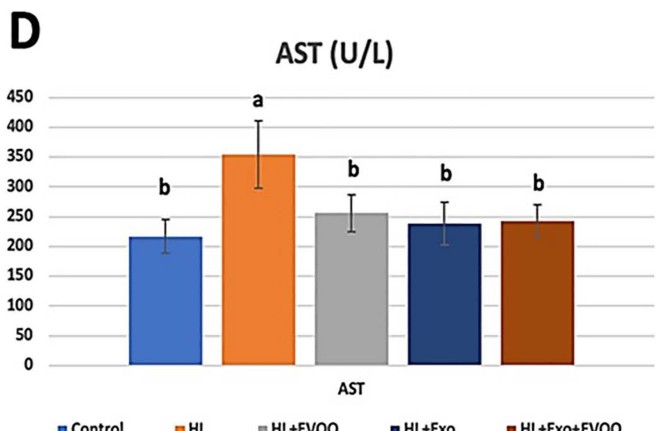

**Fig 3. Liver function parameters of different study groups. (A)** Total protein (TP, g/dL), **(B)** Albumin (Alb, g/dL), **(C)** Alanine aminotransferase (ALT, U/L), and **(D)** Aspartate aminotransferase (AST, U/L).

high fat diet fed rats which were treated with olive oil, exosomes or olive oil+exosomes and showed no significant differences when compared to the control group or to each other.

## Histopathological examination

The healthy control group showed normal histological structures of hepatic cords, sinusoids, Kupffer cells, and central veins (Fig. 4A). Liver in the hyperlipidemic rats showed intense vacuolations (indicating lipid droplets) within most hepatic sections with visible fatty changes within some hepatocytes, in addition to the presence of focal necrotic areas which are encircled by inflammatory cells infiltration (Fig. 4B, 4C). In the hyperlipidemic group treated with olive oil, the liver showed apparently normal structure of most hepatic parenchyma, central veins except for the presence of some degenerative changes in a mild number of hepatocytes (Fig. 4D). While, in the hyperlipidemic rats treated with exosomes, there were few degenerated hepatocytes within the hepatic parenchyma beside minute areas of inflammatory cells aggregates (Fig. 4E). Finally, the hyperlipidemic rats treated with both olive oil+exosomes showed the best histopathological picture, in all treated groups compared to the normal control group, with preserved hepatocytes and parenchyma architecture and in few occasions there were minute perivascular leukocytic infiltrates (Fig 4F).

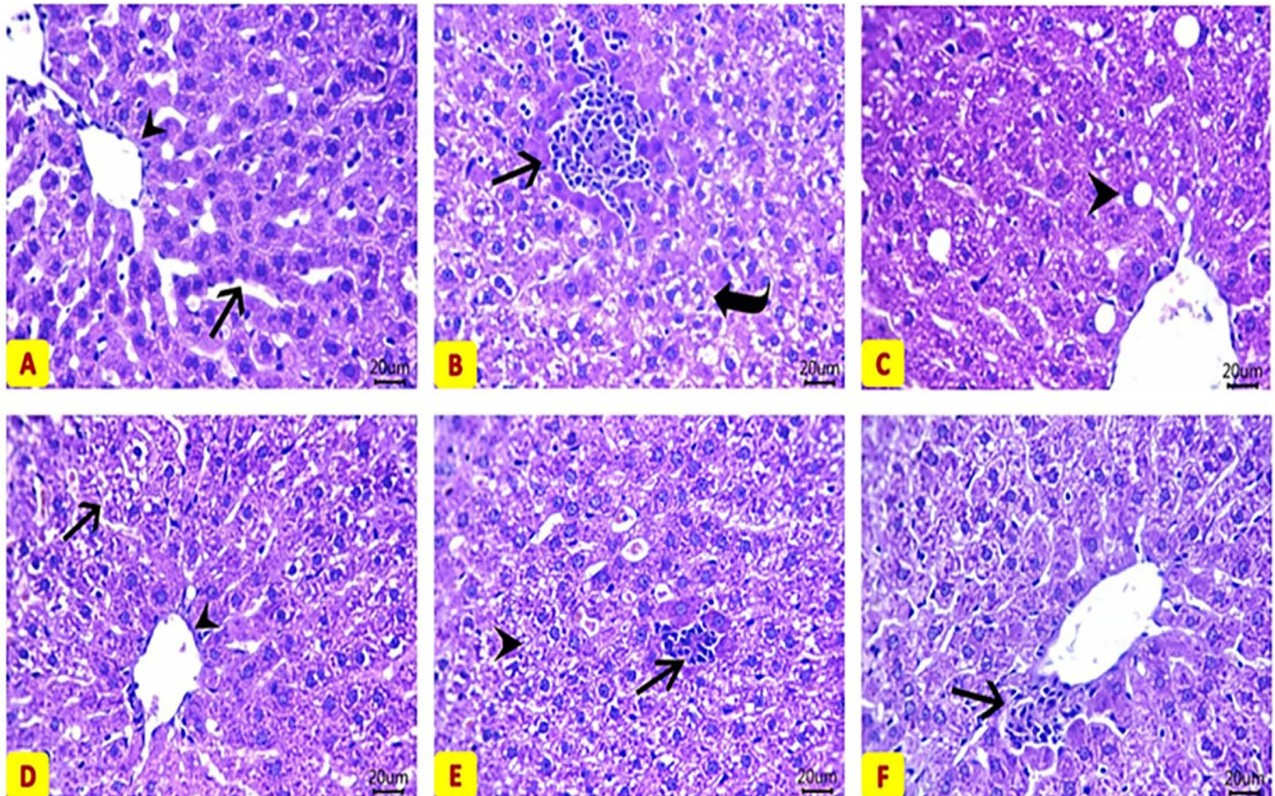

**Fig 4. Photomicrographs of H&E-stained liver sections from different study groups. (A)** Control group with normal histological structures of hepatic cords (arrow), sinusoids, Kupffur cells, and central vein (arrowhead). **(B, C)** Hyperlipidemia group showing intense vacuolations within most hepatic sections (curved arrow) with fatty changes within some cells (arrowhead). **(D)** Hyperlipidemia + Olive oil group demonstrates degenerative changes in a mild number of hepatic cells (arrow) and apparently normal central vein (arrowhead). **(E)** Hyperlipidemia + Exosomes group showing some degenerated hepatocytes (arrowhead). **(F)** Hyperlipidemia + Exosomes + Olive oil group has preserved architecture of hepatic parenchyma with minute perivascular leukocytic infiltrates (arrow).

## Gene expression analyses

The expression of genes related to the production and storage of fat inside the liver (SREBP-1c, ACC, FAS, GPAT3, SCD1, and FSP27) were all significantly (P < 0.05, Fig 5) upregulated in the liver of hyperlipidemia group.

Following olive oil or exosomes administration, their levels were significantly decreased but still higher than the control group, in case of the expression of GPAT3 and SCD1 there was no difference between olive oil or exosomes administration. In the EVOO+Exo group their levels were much reduced and were the closet to expression levels of the normal control group. In case of liver inflammation related genes (IL-1β, TNF-α, IL-6, IL-18, CCL20, and NF-κB) their expression levels were significantly (P < 0.05, Fig 6) upregulated and IL-1RN was significantly (P < 0.05, Fig 6) downregulated in the liver of hyperlipidemia group. Following olive oil or exosomes administration, the levels of IL-1β, TNF-α, IL-6, IL-18, CCL20, and NF-κB were reduced, however still higher than the normal control group and IL-1RN was increased but not as in normal control group.

The combined use of olive oil + exosomes resulted in a much better improvement in their expression levels approaching the normal control group levels. Genes related to the PPAR pathway and lipid peroxidation were significantly (P < 0.05, Fig 7) upregulated (PPARγ) or downregulated (PPARα, CPT1A, ACOX1) in the liver of hyperlipidemia group. Following olive

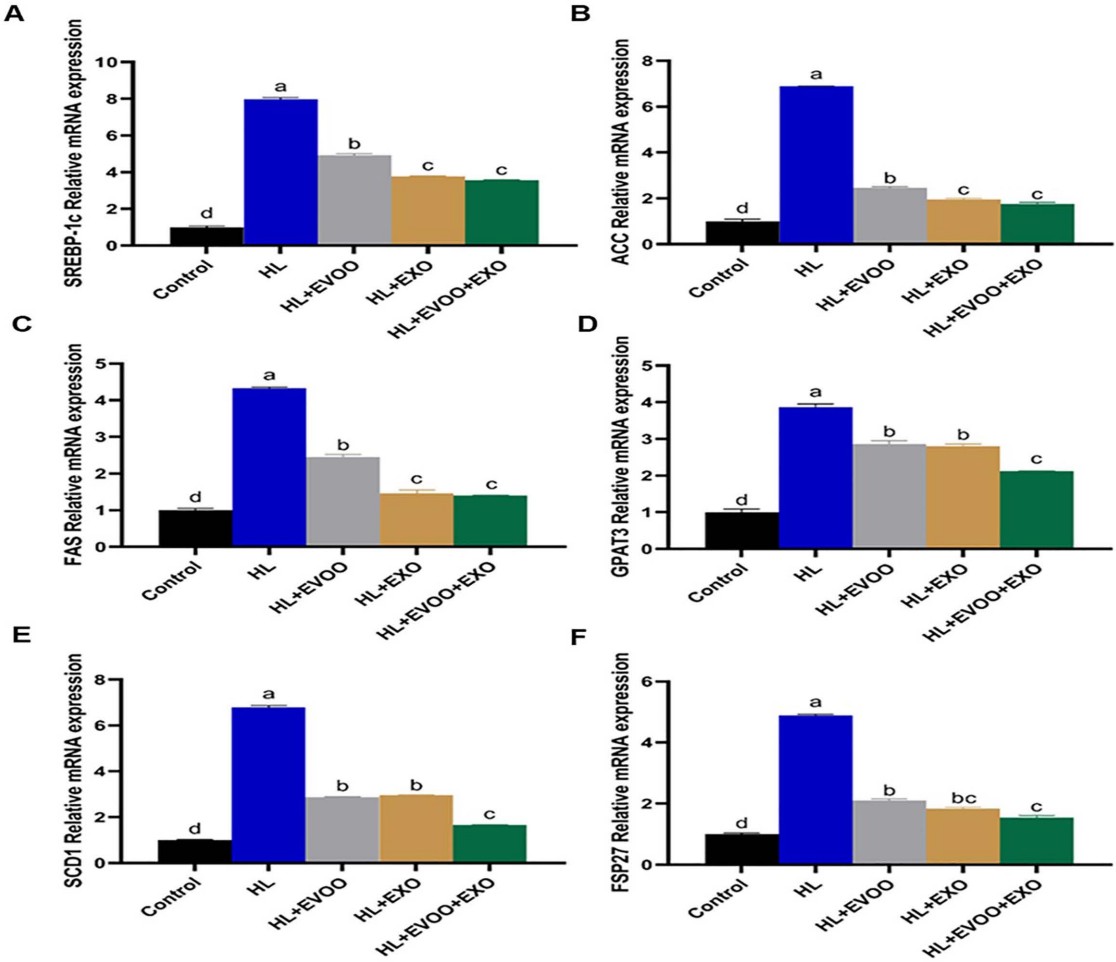

**Fig 5. The relative expression of lipogenesis related genes in the liver of different study groups. (A)** Sterol regulatory element binding transcription factor 1 (SREBP-1c), **(B)** Acetyl coenzyme A carboxylase (ACC), **(C)** Fatty acid synthase (FAS), **(D)** Glycerol-3-phosphate acyltransferase 3 (GPAT3), **(E)** Stearoyl-CoA desaturase 1 (SCD1), and **(F)** Fat-specific protein 27 (FSP27).

oil or exosomes administration their levels were restored to become nearer to normal expression levels; however, the best improvements were achieved in the combined EVOO+Exo group.

## Discussion

As a constituent of the complex pathophysiology of metabolic syndrome, hyperlipidemia caused by high-fat diet is a major challenge. Hyperlipidemia led to liver damage (i.e., NAFLD), which progresses from simple steatosis to non-alcoholic steatohepatitis (NASH), and ultimately to cirrhosis [1]. Liver steatosis sensitizes hepatocytes leading to their damage and increasing inflammation and fibrosis inside the liver [35]. These degenerative changes are also associated with increased oxidative stress, cytokine imbalance and lipid peroxidation. There is no approved treatment regimen for NAFLD and changing nutritional habits and lifestyle and even physical activity cannot provide sufficient treatment especially when liver cell damage occurs. So, we aimed to relieve liver damage, reduce liver inflammation and oxidative stress and reduce lipogenesis and correct the dyslipidemia caused by high-fat diet using a combined MSCs-Exo and EVOO approach.

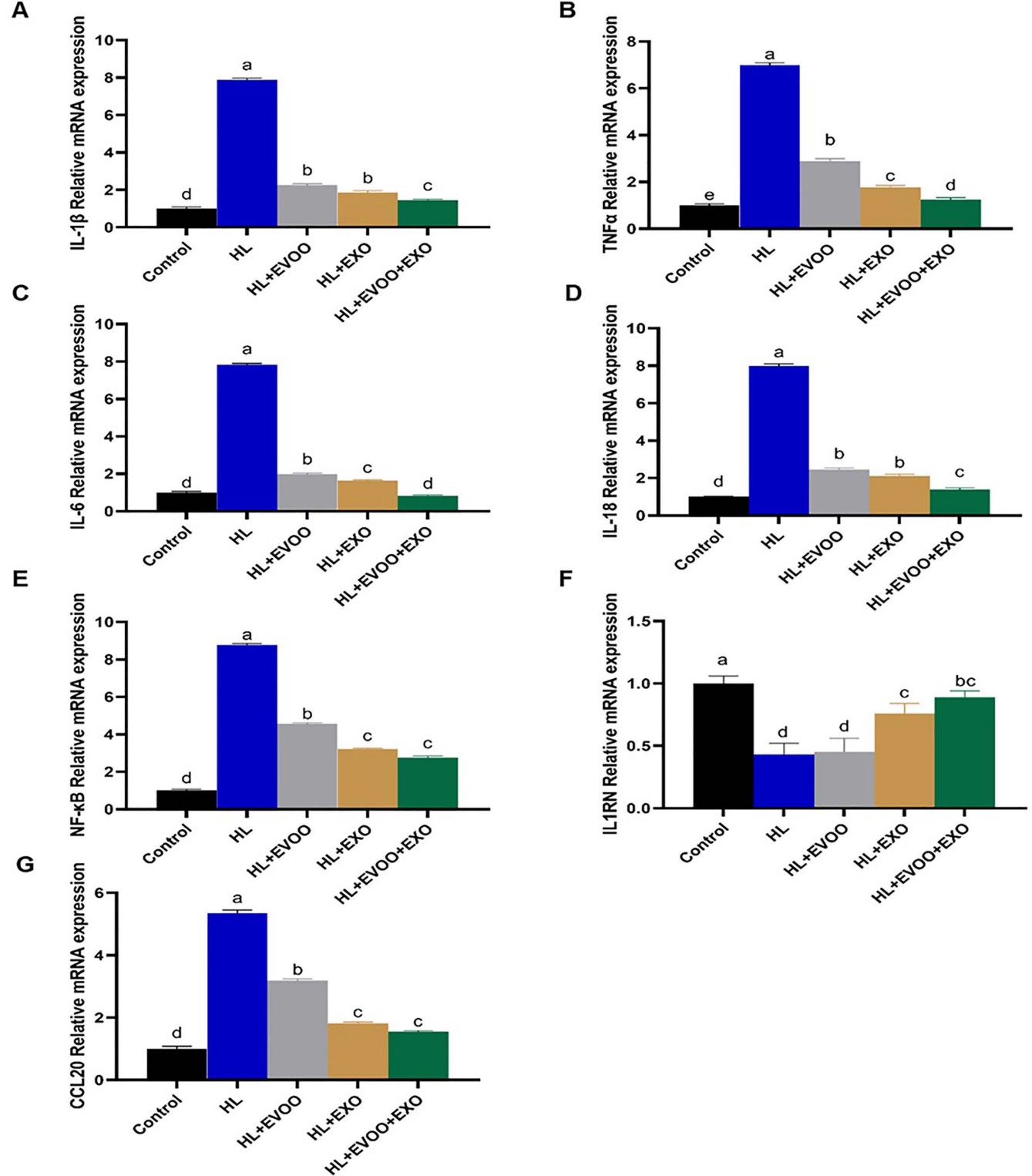

**Fig 6. The relative expression of inflammation related genes in the liver of different study groups. (A)** Interleukin-1β (IL-1β), **(B)** Tumor necrosis factor-α (TNF-α), **(C)** IL-6, **(D)** IL-18, **(E)** Nuclear factor kappa B (NF-κB), **(F)** IL-1 receptor N (IL-1RN), and **(G)** Chemokine ligand 20 (CCL20).

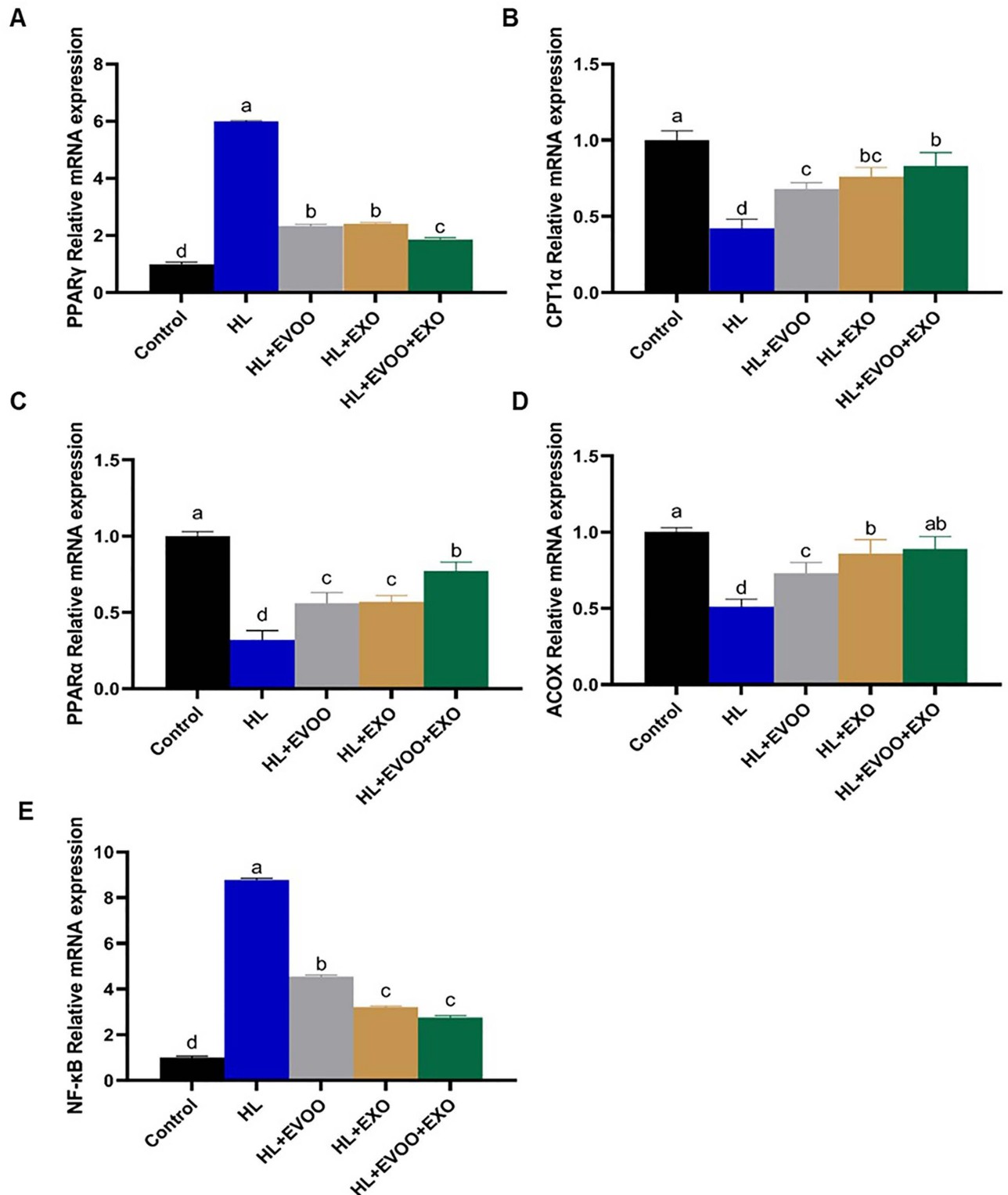

**Fig 7. The relative expression of genes related to PPAR pathway and Fatty acid oxidation pathway in the liver of different study groups. (A)** Peroxisome proliferator activated receptor alpha (PPARα), **(B)** Peroxisome proliferator activated receptor gamma (PPARγ), **(C)** Carnitine Palmitoyl transferase 1A (CPT1A), and **(D)** Acyl-CoA Oxidase 1 (ACOX1).

Following EVOO, MSCs-Exo or MSCs-Exo+EVOO administration to hyperlipidemic rats in this study, the rat's weight, liver weight and liver/rat weight percentage were all reduced and approached the normal rats values. These weight improvements were seen together with less lipid peroxidation (lower MDA) and improved oxidative parameters (higher TAC and GSH). In addition to reduction in the plasma lipid parameters (lower triglycerides, total cholesterol and LDL-C and higher HDL-C), while the liver function parameters (total protein, albumin, ALT, AST) remained near the normal levels of the control group. Histologically, the liver showed reduced degeneration, vacuolations and leukocyte infiltration indicating reduced local inflammation and reduced lipid droplets accumulation inside the liver. Olive oil alone was more powerful in reducing lipids accumulation in the liver, while exosomes were more powerful in reducing liver degeneration and inflammation. We chose the 50 μg/week i.p. regimen because it falls within the dosing ranges commonly used in rodent studies and has demonstrated biological activity in several models. Dose-response work in rats has shown significant effects with single or repeated doses starting at ~50 μg (protein) per animal, while other routes and models have used higher or lower protein-based doses depending on target tissue and administration route. The i.p. route provides a convenient, less technically demanding method than repeated intravenous injections in rodents and achieves systemic exposure (with a slower onset and lymphatic contribution compared with intravenous delivery). Nevertheless, exosome pharmacokinetics and organ tropism are influenced by route, dose, and source, so further PK/toxicology work would be required to translate toward larger animals or clinical use. Combining both MSCs-Exo+EVOO showed a much better improvement in lipid profile, antioxidant profile and reduced inflammatory and degenerative picture in the liver of hyperlipidemic rats. Better total antioxidant capacity with more GSH and much reduced lipid peroxidation were obtained in the group receiving combined MSCs-Exo and EVOO therapy, together with much higher HDL-C compared to the normal control group, indicating a better effect on reducing harmful LDL-C and increasing the beneficial HDL-C in addition to preserving the antioxidant stores in hyperlipidemic rats.

After olive oil administration in hypercholesterolemic patients, there was a noticeable reduction in total cholesterol and apolipoprotein B [14]. And, in a randomized crossover controlled trail, increasing the phenolic content of olive oil led to linear increase in HDL-C and linear reduction in LDL-C, triglycerides, oxidative stress markers [36]. In addition, recent metanalysis studies have shown the beneficial effects of olive oil on circulating metabolic factors (such as glucose and blood lipids) and highlighted its polyphenol antioxidative roles [4], and reduced cardiovascular mortality and stroke due to its MUFAs content [37]. In a different study, the intake of ozonated olive oil (which has an additional anti-inflammatory effects) with diet in Zucker (fa/fa) rats, led to reduced hepatic steatosis by inhibition of accumulation of triglycerides in the liver and suppression of inflammatory mediators [38].

We have also looked at the genes and pathways implicated in this hypolipidemic effect of combined MSCs-Exo and EVOO approach. Most importantly the genes associated with lipogenesis (*SREBP-1c, ACC, FAS*), fatty acid oxidation (*ACOX1, CPT1A, GPAT3, SCD-1* and *FSP-27),* anti-inflammatory effects (*NF-κB, IL-1β, IL-6, IL-18, IL-1RN, TNFα* and *CCL20*) and PPAR pathway (*PPARα* and *PPAR-γ)*. PPAR pathway (especially *PPAR-γ*) is critical to the progression of non-alcoholic steatohepatitis [39]. Of these genes, *GPAT3, SCD-1, IL-1RN* and *CCL20* were considered of extreme importance to the development of NAFLD and are diagnostic for the progression towards NASH and HCC [29].

ACC, as a key enzyme with FAS, which are essential for long-chain fatty acids synthesize via catalyzing the carboxylation of acetyl-CoA in a synthesis of series of malonyl CoA which generate long chain fatty acids. Moreover, hepatic Srebp-1c, has been demonstrated to selectively induce elevated expression of lipogenic genes such as ACC and SCD-1 [40]. Evidence displayed that the suppression of ACC leads to reduction in synthesis of triglyceride and its accumulation in the liver by reduced malonyl CoA activity [41]. The current study described that overexpression of lipogenic related genes in HL rats, these were prominently down regulated after a combined treatment of MSCs-Exo+EVOO indicating their synergistic role in inhibiting lipid accumulation and fatty acid biosynthesis in hepatocytes. In accordance, oleic acid rich plant oil was shown to prove that these oils beneficially control the expression of lipid metabolism associated genes [42]. Similarly to our results, the gene expression of key modulators for lipogenesis, Srebp-1c, SCD-1, FAS, and ACC, was significantly downregulated in the liver and adipose tissue of mice fed sesame oil enriched with oleic acid [43].

Impaired fatty acid oxidation (i.e., lipid catabolism and elimination) leads to excess accumulation of lipids and many metabolic disorders. Meanwhile, promoting fatty acid oxidation can aid in the prevention of metabolic disorders like dyslipidemia, obesity, NAFLD, and even NASH. Peroxisome proliferator-activated receptors (PPARs) are primary functions to sustain energy balance and lipid homeostasis, by boosting the transcription of specified targets involved in fatty acid oxidation [44]. In normal physiological condition, PPARα is highly expressed in the hepatocyte, and induce fatty acid oxidation hereby promoting lipid catabolism [45,46]. Also, PPARα activation could protect the liver from development of NAFLD and NASH by promoting the target genes correlated with fatty acid oxidation [47]. In contrast, in normal hepatocytes, PPARγ is not expressed, and its abundance is markedly elevated in fatty livers where it induces a lipogenic phenotype in both humans and rodents [48]. Furthermore, hepatic PPARγ expression augmented steatosis via up-regulating various proteins related to the uptake of lipid, TAG storage, and lipid droplets formation [49]. Additionally, carnitine palmitoyltransferase-1A (CPT1A) and Acyl-CoA oxidase (ACOX) are the two enzymes accountable for the pathways of fatty acid oxidation, while ACOX initiates the oxidation of long-chain fatty acid and CPT1A catalyzes the rate-limiting phase of fatty acid β-oxidation [50]. Notably, the lowered expression of PPARα, ACOX and CPT1A in HL rats was increased after our treatments especially when both MSCs-Exo and EVOO were combined. These findings came in line with previous work describing that treatment with a balanced diet rich in olive oil contributed to the recovery of the liver from hepatic steatosis [51]. It has been demonstrated that consumption of MUFAs declines blood TGs by stimulating fatty acid oxidation via activation of PPARα or by lowering the Srebp-1c expression and inhibiting lipogenesis [52]. Moreover, dietary MUFAs can activate PPARα, increasing lipid oxidation, and reducing resistance for insulin resulting in hepatic steatosis reduction [53]. This was achieved by decreasing activation of hepatic stellate cells by MUFAs, which are less susceptible to lipid peroxidation compared to PUFAs.

Unrefined or virgin olive oil has bioactive compounds with beneficial antioxidants action. Oleocanthal, a component found in extra virgin olive oil, is a natural anti-inflammatory compound that has a potency and profile strikingly similar to that of ibuprofen [54]. Oleic acid decreases the expression of genes involved in hepatic gluconeogenesis and lipogenesis and Srebp-1c in Zucker fatty rats [55]. Jiang *et al* have demonstrated that the pharmacological inhibition of SCD1 expression has resulted in increased fatty acid oxidation and reduced *de novo* fatty acid synthesis and thus steatosis was reduced both in hepatocyte cell line and mouse models [56]. In line with our data, MSC-Exo promotes fatty acid oxidation and lower lipogenesis in high fat diet-induced NAFLD mice [57]. This could be attributed to the protective role of MSCs-Exo+EVOO against hepatic steatosis via promoting lipid oxidation related factors, PPARα, CPT-1, and ACOX, involved in fatty acid oxidation and suppressing synthesis of fatty acid via controlling PPARα, CPT-1A, SREBP-1, and FAS expression. Additionally, exosomes attenuated systemic insulin resistance, inflammation in white adipose tissue, and hepatic steatosis in obese mice [58]. From our results, we show that the combined role of MSCs-Exo and EVOO is more potent in increasing fatty acid oxidation and reducing lipid synthesis in the liver ultimately protecting it from NAFLD progression.

The overexpression of FSP27 in HL rats contributed to TAG storage and induction of hepatosteatosis, consistent with previous work [59]. Moreover, FSP27 has been shown to decrease β-oxidation of fatty acids in adipocytes [60]. These findings indicate that FSP27 plays an important role in lipid metabolism. Remarkably, hepatic expression of FSP27 was decreased following MSCs-Exo+EVOO administration in our study suggesting a better role in ameliorating the progression of NAFLD. Glycerol-3-phosphate acyltransferase (GPAT3) is the rate-limiting enzyme in the de novo pathway of glycerolipid synthesis and plays a pivotal role in the triglyceride and phospholipid synthesis regulation. It has been shown that GPAT3 has a key role in the development of hepatic steatosis, obesity, and insulin resistance [61] Expression of GPAT3 gene, an enzyme catalyzing the initial step of fatty acid esterification resulting in synthesis of TG, was markedly upregulated among other genes involved in lipogenesis in HL rats [62]. In our study, GPAT3 was highly expressed in HL rats which indicated increased lipogenesis and TG accumulation in the hepatic tissue. In contrast its expression was remarkably reduced in HL rats treated with MSCs-Exo+EVOO. Taking together, the present findings in gene expression level

suggest that treatment with MSCs-Exo+EVOO could suppress hepatic lipogenesis and induce fatty acid oxidation in the liver tissue thus reducing NAFLD progression.

Accumulation of TG in hepatocytes in the course of hepatic steatosis resulted in an imbalance in their lipid metabolism that was followed by an increased inflammatory mediator such as cytokines, chemokines and adipocytokines that causes hepatocellular injury, inflammation and fibrosis [63]. Elevated expression of the chemokine CCL20, macrophage inflammatory protein, that serve as chemoattractant for the immune cell's infiltration to the injured hepatic tissues correlated to NASH and further play a key role in liver fibrosis in animal models [64]. In the current study, rats fed HFD suffered from steatohepatitis as they exhibited excessive inflammatory reaction at both molecular level (higher expression of CCL20, TNFα, IL-6 and IL1β) and increased levels of inflammatory histopathological areas in the liver, as previously reported in HFD fed animals [65]. Similarly, hypertrophic adipocytes produce numerous pro-inflammatory cytokines such as TNFα, IL-6 and IL1β by activation of the NF-κB pathway [66]. However, HL rats received both MSCs-Exo+EVOO displayed downregulated inflammatory markers indicating their combined impact in reducing the activation of inflammatory pathways in HL rats. Moreover, in the current study the inhibition of NF-κB explains how the combination therapy based on exosomes reduces the downstream induction of cytokine expression. IL1RN is an antagonist of the IL-1 receptor, and it is responsible for decreasing the inflammation-related activities of IL-1A and IL-1B [67]. Serum and hepatic mRNA expression levels of IL1RN have been linked as a marker for NASH [68]. Herein, the expression levels of IL1RN were restored to near normal levels in HL rats which administered both MSCs-Exo+EVOO indicating their combined beneficial role for mitigating NAFLD/NASH progression in our study rats. Recently, MSCs-Exo-based therapy was found to restore mitochondrial function and suppress inflammation and apoptosis in NAFLD [69]. Therapeutic roles of exosomes from MSCs and their function in tissue repair are widely investigated, but few reports have addressed their immunoregulatory role in NAFLD [70,71]. MSCs-Exo can downregulate lipid metabolism-related gene expression and reduce lipid deposition in NAFLD rats [72]. Also, it was described that MSCs-Exo promote fatty acid oxidation and reduce lipogenesis and expression of inflammatory factors TNF-α, IL-1, and IL-6 in the liver in oleic–palmitic acid-treated hepatic cells and HFD-induced NAFLD mice [56]. The decrease in inflammatory reaction of olive oil receiving rats fed HFD was attributed to its higher content from MUFA (oleic acid) which reduced oxidized LDL [73], LDL cholesterol and the concentration of TG without the concurrent reduction in HDL these results also confirmed by previous work [74]. Additional effects of EVOO beyond its MUFA composition relate to its polyphenols. Polyphenols present in EVOO, such as oleuropin, hydroxytyrosol, tyrosol and caffeic acid, have an important antioxidant and anti-inflammatory effect [40]. The principal mechanisms of action of olive oil include the decrease in NF-κB activation and decrease in LDL oxidation [51]. The prominent impact of MSCs-Exo on inflammatory response in HFD rats was attributed to their role in inhibiting macrophage inflammatory response indicating their role in reducing obesity-associated inflammation [74]. In addition to high MUFA content, EVOO contains a considerable amount of antioxidants, α-tocopherol and phytochemicals that inhibit inflammation, insulin resistance, mitochondrial dysfunction, endoplasmic reticulum stress, that lead to the resolution or prevention of liver injury [51].

## Conclusion

The combined effect of exosomes derived from MSCs, and extra virgin olive oil had a much improved effect on the reduction of lipid accumulation in the liver and reduction of the inflammatory mediators and local inflammation in the liver. Together with improved fatty acid oxidation, improved antioxidant stores, reduced oxidative stress and reduction of gene expression of the most prominent hyperlipidemia associated markers. All this indicates that the combined effect of exosomes derived from MSCs, and extra virgin olive oil are promising in combating the worldwide spread of hyperlipidemia and its associated diseases such as NAFLD and NASH. Further research is required to completely elucidate the exact mechanisms of this synergistic combined effect.

## Limitations of the study

One limitation of this study is that the specific chemical composition of the extra virgin olive oil used was not analytically determined. While the EVOO met standard quality criteria and was sourced from a certified supplier, future studies should include detailed compositional profiling to better link specific compounds to biological effects. Also, further dose-response studies would be beneficial in future research.

## Author contributions

**Conceptualization:** Bassam F. Alowaiesh, Ayman A. Saleh, Haifa A. S. Alhaithloul.

**Data curation:** Doaa Ibrahim, Haifa A. S. Alhaithloul.

**Formal analysis:** Ayman A. Saleh, Ahmed Abdelfattah-Hassan.

**Funding acquisition:** Haifa A. S. Alhaithloul.

**Investigation:** Bassam F. Alowaiesh, Doaa Ibrahim, Abdulsalam A. M. Alkhaldi, Ahmed Abdelfattah-Hassan.

**Methodology:** Bassam F. Alowaiesh, Doaa Ibrahim, Abdulsalam A. M. Alkhaldi, Ahmed Abdelfattah-Hassan.

**Resources:** Bassam F. Alowaiesh, Doaa Ibrahim, Abdulsalam A. M. Alkhaldi, Ahmed Abdelfattah-Hassan.

**Supervision:** Bassam F. Alowaiesh, Ayman A. Saleh, Haifa A. S. Alhaithloul, Abdulsalam A. M. Alkhaldi.

**Visualization:** Abdulsalam A. M. Alkhaldi.

**Writing – original draft:** Doaa Ibrahim, Ayman A. Saleh, Haifa A. S. Alhaithloul, Abdulsalam A. M. Alkhaldi, Ahmed Abdelfattah-Hassan.

**Writing – review & editing:** Doaa Ibrahim, Ayman A. Saleh, Haifa A. S. Alhaithloul, Abdulsalam A. M. Alkhaldi, Ahmed Abdelfattah-Hassan.

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
