## [Decision Letter · Decision Letter 0]

23 Apr 2025

Dear Dr. falawish,

Thank you for submitting your manuscript to PLOS ONE. After careful consideration, we feel that it has merit but does not fully meet PLOS ONE’s publication criteria as it currently stands. Therefore, we invite you to submit a revised version of the manuscript that addresses the points raised during the review process.

We look forward to receiving your revised manuscript.

Kind regards,

Hualin Wang

Academic Editor

PLOS ONE

 [This work was funded by the Deanship of Scientific Research in cooperation with Olive Research Center at Jouf University under Grant Number (DSR2022-RG-0164).]. 

6. PLOS requires an ORCID iD for the corresponding author in Editorial Manager on papers submitted after December 6th, 2016. Please ensure that you have an ORCID iD and that it is validated in Editorial Manager. To do this, go to ‘Update my Information’ (in the upper left-hand corner of the main menu), and click on the Fetch/Validate link next to the ORCID field. This will take you to the ORCID site and allow you to create a new iD or authenticate a pre-existing iD in Editorial Manager.

7. Please include a separate caption for each figure in your manuscript.

8. Please include a copy of Table 1-2 which you refer to in your text on page 9.

Additional Editor Comments (if provided):

Reviewers' comments:

Reviewer's Responses to Questions

**Comments to the Author**

1. Is the manuscript technically sound, and do the data support the conclusions?

Reviewer #1: Yes

Reviewer #2: Partly

2. Has the statistical analysis been performed appropriately and rigorously?

Reviewer #1: Yes

Reviewer #2: No

3. Have the authors made all data underlying the findings in their manuscript fully available?

Reviewer #1: Yes

Reviewer #2: Yes

4. Is the manuscript presented in an intelligible fashion and written in standard English?

Reviewer #1: Yes

Reviewer #2: No

Reviewer #1: This manuscript is " Exosomes combined with extra virgin olive oil reduces lipogenesis, oxidative stress, and inflammation in non-alcoholic fatty liver disease model ". The topic of this manuscript is interesting and suitable for publication in the journal. However, some points need explanation, which are mentioned below:

1. Do not use abbreviations in the keywords section.

2. In the introduction, you explained that extra virgin oil has important nutritional effects. Given that impurities in the oil are not removed, please also explain if it has anti-nutritional properties.

3. Could the special properties of extra virgin oil be due to compounds other than fatty acids and total antioxidant?

4. Did you determine the composition of extra virgin oil؟

5. It is not necessary to use "showing" at the beginning of the names of arguments and forms.

Reviewer #2: This study used exosomes derived from BM-MSCs, combined with extra virgin olive oil (EVOO), to treat high fat diet fed rat. The authors reported that administration of olive oil, or exosomes, significantly improved hyperlipidemia in rats, including rat’s body weight, plasma lipid parameters, liver parameters, and lipogenesis-related gene expression; however, the combined EVOO+Exo group had the best improvements. This study is the first to combine exosomes with olive oil for the treatment of NAFLD.

Major issues:

1. Statistical analysis: in the figure legend, “Bars with different letters show statistical significance between groups (P<0.05) using Tukey's HSD.” What do different letters exactly mean? a, b, c? Please provide all the p values.

2. Related to issue 1. From the bar charts shown in certain figures, the efficacy advantage of combined administration seems to be not significant when compared with olive oil or exosomes.

3. In the experimental design, the hyperlipidemia group was treated with 50μg of exosomes derived from BM-MSCs once a week by intraperitoneal injection. Why did the author use i.p., instead of i.v.? In addition, how to determine the dosage of exosomes?

4. The phenotype of NAFLD in the rat model is not obvious. Is there any index of NAFLD in rats fed with high fat diet? Or is the disease model just hyperlipidemia?

Minor issues:

1. Figure 4: label arrow, arrowhead correctly.

2. Make sure the references are appropriate.

3. Some sentences are repetitive, such as page 15, line 8-10.

**Do you want your identity to be public for this peer review?** For information about this choice, including consent withdrawal, please see our Privacy Policy

Reviewer #1: No

Reviewer #2: **Yes: ** chen wang

---

## [Author Response · Author response to Decision Letter 1]

25 May 2025

Dear Reviewers,

Thank you for the review of our manuscript (PONE-D-25-10509) titled: “Exosomes combined with extra virgin olive oil reduces lipogenesis, oxidative stress, and inflammation in non-alcoholic fatty liver disease model”. The whole manuscript has been revised carefully, and we have considered all your comments. These revisions are highlighted in the manuscript by track changes and are summarized in the attached pdf file named responses to reviewers.

We hope the changes that have been made were appropriate and the manuscript can now be accepted for publication.

---

## [Decision Letter · Decision Letter 1]

2 Jul 2025

Dear Dr. Falawish,

Thank you for submitting your manuscript to PLOS ONE. After careful consideration, we feel that it has merit but does not fully meet PLOS ONE’s publication criteria as it currently stands. Therefore, we invite you to submit a revised version of the manuscript that addresses the points raised during the review process.

We look forward to receiving your revised manuscript.

Kind regards,

Hualin Wang

Academic Editor

PLOS ONE

Additional Editor Comments:

The authors should explain the dosage of exosomes and the statistical analysis carefully.

Reviewers' comments:

Reviewer's Responses to Questions

**Comments to the Author**

Reviewer #2: (No Response)

2. Is the manuscript technically sound, and do the data support the conclusions?

Reviewer #2: Yes

3. Has the statistical analysis been performed appropriately and rigorously?

Reviewer #2: I Don't Know

4. Have the authors made all data underlying the findings in their manuscript fully available?

Reviewer #2: Yes

5. Is the manuscript presented in an intelligible fashion and written in standard English?

Reviewer #2: Yes

Reviewer #2: Related to comment 3:

About the injection method and dosage of exosomes, please add references to support this and discuss.

**Do you want your identity to be public for this peer review?** For information about this choice, including consent withdrawal, please see our Privacy Policy

Reviewer #2: No

---

## [Decision Letter · Decision Letter 2]

17 Sep 2025

Exosomes combined with extra virgin olive oil reduces lipogenesis, oxidative stress, and inflammation in non-alcoholic fatty liver disease model

PONE-D-25-10509R2

Dear Dr. falawish,

We’re pleased to inform you that your manuscript has been judged scientifically suitable for publication and will be formally accepted for publication once it meets all outstanding technical requirements.

Kind regards,

Hualin Wang

Academic Editor

PLOS ONE

Additional Editor Comments (optional):

Please check page 57, as the reviewer 2 mentioned: "When explaining exosome dosage and administration, there are some repetitive statements. Please revise.".

Reviewer #2:

Reviewers' comments:

Reviewer's Responses to Questions

**Comments to the Author**

Reviewer #2: All comments have been addressed

2. Is the manuscript technically sound, and do the data support the conclusions?

Reviewer #2: Yes

3. Has the statistical analysis been performed appropriately and rigorously?

Reviewer #2: Yes

4. Have the authors made all data underlying the findings in their manuscript fully available?

Reviewer #2: Yes

5. Is the manuscript presented in an intelligible fashion and written in standard English?

Reviewer #2: Yes

Reviewer #2: Page 57: When explaining exosome dosage and administration, there are some repetitive statements. Please revise.

**Do you want your identity to be public for this peer review?** For information about this choice, including consent withdrawal, please see our Privacy Policy

Reviewer #2: No

---

## [Editor Report · Acceptance letter]

PONE-D-25-10509R2

PLOS ONE

Dear Dr. falawish,

I'm pleased to inform you that your manuscript has been deemed suitable for publication in PLOS ONE. Congratulations! Your manuscript is now being handed over to our production team.

Kind regards,

on behalf of

Dr. Hualin Wang

Academic Editor

PLOS ONE